Efficacy and optimal dosage of various exercises for migraine: a multilevel network and dose-response meta-analysis

Xie Jingyi 1
Lin Yupeng 2
Wang Bin 19107138519@163.com 1
1 Department of Physical Education, Central China Normal University , Wuhan , China
2 Department of Physical Education, China University of Mining Technology , Beijing , China
Farhan Faiza
Electronic publication date: 2025 Oct 20
Publication date: 2025
Volume: 13
Electronic Location ID: e20254
Received 2025 Jul 30; Accepted 2025 Sep 26
Copyright: ©2025 Xie et al.
Copyright year: 2025
Copyright holder: Xie et al.
License: This is an open access article distributed under the terms of the Creative Commons Attribution License, which permits unrestricted use, distribution, reproduction and adaptation in any medium and for any purpose provided that it is properly attributed. For attribution, the original author(s), title, publication source (PeerJ) and either DOI or URL of the article must be cited.
License URL: https://creativecommons.org/licenses/by/4.0/

Keywords: Exercise, Migraine

Funding: The authors received no funding for this work.

==============================
Background

To elucidate the relative efficacy of diverse exercise modalities for migraine and quantify the optimal therapeutic dosage.

Method

A systematic search was conducted across four electronic databases from their inception to May 2025. Two reviewers independently performed data extraction and risk of bias assessment. A multilevel network meta-analysis (ML-NMA) integrated with a dose-response analysis was employed to comprehensively compare these interventions.

Results

Twenty-seven randomized controlled trials (RCTs) (n = 1,611) were included. The most effective interventions were combined aerobic+resistance exercise (g =  − 1.85, 95% credible interval (CrI): [−2.53 to −1.18]; surface under the cumulative ranking curve (SUCRA) = 0.91), followed by resistance exercise (g =  − 1.45, 95% CrI [−1.79 to −1.10]; SUCRA = 0.81), yoga (g =  − 0.35, 95% CrI [−0.63 to −0.06]; SUCRA = 0.49), and tai chi (g =  − 0.48, 95% CrI [−0.91 to −0.05]; SUCRA = 0.46). The dose-response analysis identified an optimal therapeutic window of 300-600 metabolic equivalent of task (MET)-min/week, an intensity of 4.5–5.5 METs, and a duration of 8–10 weeks. The overall certainty of evidence was rated from very low to low.

Conclusion

Combined aerobic+resistance exercise, resistance exercise, tai chi, and yoga represent promising therapeutic options for migraine. The optimal dose was identified as approximately 70–135 minutes of moderate-intensity or 45–90 minutes of vigorous-intensity activity weekly, for 8–10 weeks. These findings, however, must be interpreted with caution due to the low quality of the underlying evidence.

Introduction

The rising global prevalence of migraine, a common and burdensome neurological disorder, presents a significant public health challenge (Steiner & Stovner, 2023; Silberstein, 2004; Dalessio, 1990; Montagna, 2008; Dong et al., 2025). Among various therapeutic strategies, exercise has emerged as a promising non-pharmacological intervention, owing to its favorable safety profile, accessibility, and cost-effectiveness (Diener et al., 2015; Wells et al., 2011; Varkey et al., 2011).

However, the optimal exercise prescription for migraine remains poorly defined. Previous network meta-analyses (NMA) have yielded inconsistent findings and failed to provide clear clinical guidance, largely because conventional statistical methods cannot adequately account for the wide variation in exercise dosages across studies (Reina-Varona et al., 2024; Woldeamanuel & Oliveira, 2022; Mills, Thorlund & Ioannidis, 2013; Caldwell, Ades & Higgins, 2005). By aggregating disparate doses, these analyses risk masking crucial dose–response relationships, leaving the therapeutic impact of specific exercise parameters largely uncharacterized.

To address this critical gap, this study employs a multilevel network meta-analysis (ML-NMA). This advanced approach enables a systematic comparison of the relative efficacy of different exercise modalities while simultaneously modeling their dose–response relationships. Our goal is to establish a robust evidence base to inform the clinical prescription of exercise in migraine management.

Methods

Protocol and registration

We conducted this ML-NMA following the guidelines of the Cochrane Handbook for Systematic Reviews of Interventions and the PRISMA extension statement for Network Meta-Analyses. The study protocol was prospectively registered in PROSPERO (CRD42025633072) (Higgins et al., 2024).

Search strategy and study selection

We conducted a comprehensive systematic search of four electronic databases (PubMed, Embase, Web of Science, and the Cochrane Library) from their inception to May 2025, with no language restrictions. Our literature search strategy was twofold to ensure comprehensive coverage. First, we conducted electronic searches using a combination of Medical Subject Headings (MeSH) and free-text keywords. Second, this was complemented by hand-searching the bibliographies of identified systematic reviews and meta-analyses for any additional studies (Supplementary S1).

Eligibility and exclusion criteria

Eligibility criteria were defined using the Population, Intervention, Comparator, Outcomes, and Study (PICOS) framework.

(i) Population: We included studies enrolling adult patients whose migraine diagnosis was established in accordance with the International Classification of Headache Disorders, 3rd edition (ICHD-3) (Olesen, 2018). This required confirmation that patients met the specific diagnostic criteria for migraine, covering aspects like attack frequency, duration, characteristics, and associated symptoms. (ii) Interventions: Eligible interventions consisted of structured exercise modalities, such as yoga, aerobic exercise, resistance training, tai chi, stretching, high-intensity interval training (HIIT), and their combinations. (iii) Comparators: Eligible comparator groups included no-intervention controls, usual care, or waitlist controls. (iv) Outcomes: Key endpoints were validated measures of pain intensity, such as the Visual Analogue Scale (VAS), Numeric Rating Scale (NRS), and Total Pain Rating Index (T-PRI), and of migraine-specific disability, including the Headache Impact Test-6 (HIT-6) and Migraine Disability Assessment Scale (MIDAS). (v) Study design: Only randomized controlled trials (RCTs) were included in this analysis.

We excluded studies with crossover designs, as well as conference abstracts, protocols, and systematic reviews. Studies were also excluded if they provided insufficient data for extraction or if the required data were inaccessible upon author request.

Data extraction

Following deduplication with EndNote (version 20), study selection was conducted in duplicate. Two reviewers (JX, YL) independently performed an initial screening of titles and abstracts, followed by a full-text eligibility assessment of all potentially relevant records. Any discrepancies encountered during screening or extraction were resolved through discussion or adjudication by a third senior author (BW). Extracted variables included first author, publication year, participant demographics (age, sex, sample size), detailed intervention parameters (type, duration, frequency, intensity), and all relevant pain intensity outcomes. To facilitate the dose–response analysis, each intervention arm within a single study that featured a unique exercise dosage was treated as an independent node in the network.

Calculation of the exercise dose

To standardize the exercise dose, we calculated the total volume in MET-minutes per week. This was the product of session frequency, duration (main component only, excluding warm-up/cool-down), and the corresponding MET value. METs were assigned based on the 2024 Compendium of Physical Activities and ACSM guidelines (Herrmann et al., 2024; Glass, 2024). For studies not reporting session duration, the value was imputed using the mean from comparable included studies (Liang et al., 2024).

Risk of bias and certainty of evidence

The methodological quality of each RCT was assessed via the Cochrane Risk of Bias 2.0 (RoB 2.0) tool, which evaluates bias across five domains (randomization process, deviations from interventions, missing data, outcome measurement, and selective reporting). Each domain was classified as having a ‘low risk’, ‘high risk’, or ‘some concerns’ (Sterne et al., 2019).

For the synthesis of evidence, the CINeMA framework was employed to determine the certainty of each network estimate. This process involved rating six key aspects—within-study bias, reporting bias, indirectness, imprecision, heterogeneity, and incoherence—to arrive at a final confidence level of ‘high’, ‘moderate’, ‘low’, or ‘very low’ (Nikolakopoulou et al., 2020). All quality and certainty assessments were carried out in duplicate by two independent reviewers (JX, YL), with a third reviewer (BW) mediating any unresolved conflicts.

Statistical analysis

All statistical analyses were conducted in R (version 4.4.1), where a two-sided p-value <0.05 was considered statistically significant.

Effect size calculation: To quantify the effect size for each comparison, we selected the standardized mean difference (SMD), specifically calculated as Hedges’ g to correct for potential small-sample bias. This computation was performed using the esc package, and the resulting effect magnitudes were classified as small (g ≥ 0.2), moderate (g ≥ 0.5), or large (g ≥ 0.8) based on established conventions (Hedges & Olkin, 1985).

Network consistency: We first assessed the core assumption of transitivity by clinically and methodologically comparing studies across treatment comparisons. Statistical consistency was then evaluated globally using a design-by-treatment interaction model and locally using the node-splitting method within the netmeta package (Dias et al., 2010; Higgins et al., 2012).

ML-NMA: We implemented an arm-based Bayesian multilevel model using the brms package. This hierarchical structure effectively managed the statistical dependency arising from multiple effect sizes (e.g., different outcome scales) clustered within a single study arm. We employed weakly informative priors to ensure model stability and confirmed model convergence by monitoring the potential scale reduction factor (PSRF), ensuring all values were below 1.05 (Lin et al., 2017; Brooks & Gelman, 1998). Treatment effects are reported as Hedges’ g with 95% credible intervals (CrI). Intervention rankings were determined using surface under the cumulative ranking (SUCRA) values.

Dose-response analysis: First, we fitted a Bayesian multilevel model with natural splines (four knots) using the brms package, regressing the study-specific effect sizes (Hedges’ g) on the corresponding exercise volumes. Second, based on the posterior predictive distribution of the model, we generated a dose–response curve by estimating the expected Hedges’ g and its 95% CrI for a fine grid of exercise volumes. The optimal dose was defined as the point estimate yielding the highest effect, while the optimal range was the interval where the lower bound of the 95% CrI for the effect size was greatest.

Publication bias and clinical significance: Publication bias was assessed via funnel plot symmetry and Egger’s regression test. The minimal clinically important difference (MCID) was calculated (SMD (g)–0.4 * sd (g)) to find the probability of an intervention’s effect surpassing this value (Watt et al., 2021).

Acceptability analysis: Dropout rates served as a proxy for intervention acceptability. Accordingly, we calculated the odds ratios (OR) for the dropout rate of each intervention versus the control group using the netmeta package.

Result

Study selection

Our systematic search identified a total of 10,254 records. After eliminating duplicates, the titles and abstracts of the remaining 3,411 articles were assessed for eligibility. This initial screening phase culminated in the selection of 27 studies for the final ML-NMA (Varkey et al., 2011; Alipouri et al., 2023; Butt et al., 2022; Darabaneanu et al., 2011; Eslami et al., 2021; John et al., 2007; Krøll et al., 2018; Aslani et al., 2022; Bond et al., 2018; Boroujeni et al., 2015; Johnson et al., 2025; Niu et al., 2025; Kisan et al., 2014; Kumar et al., 2020; Kumari et al., 2022; Lemstra, Stewart & Olszynski, 2002; Matin, Taghian & Chitsaz, 2022; Mehta et al., 2021; Narin et al., 2003; Oliveira et al., 2019; Baykan Çopuroğlu & Çopuroğlu, 2024; Dündar et al., 2024; Sun et al., 2022; Rahimi et al., 2023; Fernando Prieto Peres, Prieto Peres Mercante & Belitardo de Oliveira, 2019; Xie et al., 2022; Kaushal et al., 2023) (Fig. 1).

Figure 1 Literature review flowchart.

Description of clinical trials

The pooled data were drawn from 1,611 participants (80.0% female; n = 1,289), with study-level mean ages varying from 20 to 60 years. The therapeutic strategies were categorized into seven types of exercise: aerobic (n = 14 studies), yoga (n = 9), stretching (n = 2), tai chi (n = 1), resistance (n = 2), combined aerobic/resistance (n = 1), and HIIT (n = 2). The discrepancy between the sum of studies per category and the total number of unique reports is attributable to the inclusion of multi-arm studies that compared several interventions simultaneously Supplementary S2.

Risk of bias and quality of evidence

An assessment of the risk of bias for the 27 included RCTs identified seven studies at high risk, 16 with some concerns, and four at low risk (Supplementary S3). The certainty of evidence for pairwise interventional comparisons, evaluated via the CINeMA framework, was graded as low to very low for all comparisons (Supplementary S4).

Multilevel network meta-analysis

A comparison of the main characteristics across the included studies showed no clear systematic differences between studies involved in different comparisons. The global design-by-treatment interaction model yielded results that were not statistically significant (χ2 = 93.15, df = 76, P = 0.061), indicating that the transitivity assumption holds. Additionally, the node-splitting analysis detected no significant inconsistencies within the networks (Supplementary S5). No compelling evidence was found to suggest a violation of the transitivity assumption.

The network evidence plot and forest plot are presented in Fig. 2. Based on SUCRA values, which provide a probabilistic ranking of interventions, the most effective interventions were: Combined aerobic+resistance exercise (g = −1.85, 95% CrI [−2.53 to −1.18]; SUCRA = 0.91), followed by resistance exercise (g = −1.45, 95% CrI [−1.79 to −1.10]; SUCRA = 0.81), yoga (g = −0.35, 95% CrI [−0.63 to −0.06]; SUCRA = 0.49), and tai chi (g = −0.48, 95% CrI [−0.91 to −0.05]; SUCRA = 0.46). The effects of HIIT, aerobic exercise, and stretch exercise, were not statistically significant compared to controls (Fig. 3).

Figure 2 Network diagrams depicting the direct and indirect comparisons for the network meta-analyses.

The size of the nodes represents the number of participants in each intervention. The connections between the nodes represent a direct comparison of different interventions, and their thickness indicates the amount of direct evidence.

Figure 3 Network meta-analysis forest plot of various exercise interventions on migraine pain intensity.

Funnel plot asymmetry and significant Egger’s regression test results (P < 0.001) suggested a potential publication bias (Supplementary S6). Regarding acceptability, most exercise interventions did not show a statistically significant difference in dropout rates compared to controls. However, tai chi demonstrated a significantly lower dropout rate (OR = 0.23) (Supplementary S7).

Moderation by participant characteristics

The results indicated that the gender composition of the study samples did not significantly moderate the effect of the exercise interventions. There was little evidence suggesting that mean participant age substantially moderates the effect of exercise on migraine improvement. Intervention supervision or funding status did not significantly moderate the effect on migraine outcomes (Supplementary S8).

Dose–response analysis

The dose–response analysis explored the relationship between key exercise parameters (weekly dose, intensity, and intervention duration) and migraine outcomes.

First, a non-linear relationship was identified between the weekly exercise dose and treatment effect. While exercise proved effective across all tested doses, the range of 300 to 600 MET-min/week emerged as an optimal therapeutic window. This range offered a high probability (>60%) of achieving a minimal clinically important difference (MCID, g <  − 0.4) and was supported by the highest statistical certainty (i.e., the narrowest 95% CrI), making it a practical therapeutic target. Further analysis of the dose components revealed that an exercise intensity of 4.5−5.5 METs and an intervention duration of 8-10 weeks were associated with the most robust and clinically significant effects, also supported by high statistical precision in our models. In contrast, the treatment effect remained stable across different follow-up periods, suggesting no significant decline over time (Fig. 4 and Supplementary S9).

Figure 4 Dose–response relationship between overall exercise dosage and migraine.

Discussion

Principal findings

Our network meta-analysis establishes that combined aerobic+resistance training, resistance training alone, tai chi, and yoga are effective interventions for migraine. The primary contribution of this work, however, is the delineation of an evidence-based dosage framework from our dose–response analysis. We identified a non-linear relationship where a therapeutic window of 300–600 MET-min/week, an intensity of 4.5–5.5 METs, and a duration of 8–10 weeks defines an optimal balance between clinical efficacy and statistical certainty. This key finding challenges the notion that simply maximizing exercise volume is the optimal strategy.

Comparison with existing evidence

Our finding that a combined aerobic and resistance regimen yields robust efficacy corroborates the broader literature. While a previous meta-analysis suggested yoga as the optimal intervention (Reina-Varona et al., 2024), our analysis positions combined training as superior, despite confirming yoga’s effectiveness. The benefits of yoga are likely mediated by its mind-body regulatory mechanisms (Wu et al., 2022); however, it provides less intense physiological stimuli for systemic cardiovascular and muscular adaptation compared to combined training. The latter offers a more comprehensive stimulus, promoting adaptations across cardiorespiratory, muscular, and metabolic domains to collectively enhance resilience against migraine triggers (Schumann et al., 2022; Niu et al., 2024). Regarding aerobic exercise, our findings support previous meta-analyses establishing its benefits (Lemmens et al., 2019; La Touche et al., 2020). However, its relative advantage was less pronounced within our comparative NMA framework, likely because it offers less targeted muscle stimulation than resistance training and less mind-body integration than practices like yoga and tai chi (American College of Sports Medicine, 2021).

Notably, our analysis underscores the considerable potential of resistance exercise, both as a standalone modality and as part of a combined regimen (Woldeamanuel & Oliveira, 2022). This efficacy may stem from unique physiological mechanisms, including targeted improvements in musculoskeletal function, distinct neuroendocrine responses, or specific impacts on central pain modulation pathways relevant to migraine pathophysiology (Naugle, Fillingim & Riley 3rd, 2012).

Finally, our identification of an optimal dosage window (300–600 MET-min/week) offers a more nuanced perspective than a simplistic “more is better” paradigm. This volume, equivalent to 70–135 min of moderate-intensity or 45–90 min of vigorous-intensity activity weekly (Bull et al., 2020), appears to balance efficacy with feasibility. The existence of an optimal range, rather than a linear dose–response, likely reflects a complex interplay of factors. Physiologically, excessive exercise can act as a stressor, potentially triggering headaches and creating an “inverted-U” response curve (Amin et al., 2018). Behaviorally, overly demanding regimens can undermine adherence, reducing long-term effectiveness. Biologically, the benefits may be subject to a “ceiling effect”, where additional volume yields diminishing marginal returns. The optimal duration of 8–10 weeks likely reflects the time required for these physiological adaptations to consolidate and for new behavioral habits to form (Lally et al., 2010).

Clinical implications and practical guidance

Our findings provide actionable guidance for clinicians. When managing patients with migraine, clinicians can recommend several evidence-supported modalities, including combined aerobic and resistance training, resistance exercise alone, tai chi, and yoga. For exercise prescription, patients can be guided toward a weekly target of 70–135 min of moderate-intensity activity (e.g., brisk walking) or 45–90 min of vigorous-intensity activity (e.g., jogging). This volume aligns with the dose–response profile associated with favorable outcomes in our analysis. Furthermore, promoting adherence for at least 8–10 weeks is critical, as this duration was linked to significant therapeutic effects. Crucially, all exercise recommendations must be tailored to the individual’s capabilities, preferences, and contraindications.

Broader perspectives: comorbidity and combined interventions

A significant clinical consideration, not fully addressed by the included studies, is the frequent comorbidity of migraine and tension-type headache (TTH). This coexistence is common in clinical practice, often resulting in a more complex presentation, greater headache burden, and increased disability (Ashina et al., 2021). While our analysis focused on migraine, the physiological and psychological benefits of exercise—such as stress reduction and improved musculoskeletal function—are also relevant to TTH management. This mechanistic overlap suggests the need for therapeutic strategies that address both the physiological and psychological components of headache. Consequently, the integration of exercise with established psychological therapies warrants consideration.

Among psychological interventions, cognitive behavioral therapy (CBT) is a well-established, evidence-based option for chronic pain, including both migraine and TTH. CBT functions by helping patients modify maladaptive cognitions (e.g., pain catastrophizing), enhance coping strategies, and manage stress—a common trigger for both headache types (GBD 2016 Headache Collaborators, 2018).

A multimodal approach combining exercise and CBT may offer synergistic benefits. Exercise addresses the physiological underpinnings (e.g., improving physical conditioning, releasing endorphins), while CBT targets the psychological drivers of pain perception and behavior. This dual-pronged strategy could be particularly effective for the substantial patient population with comorbid migraine and TTH, offering a more holistic and potent therapeutic model. Therefore, future high-quality RCTs are needed to evaluate the efficacy of such integrated approaches against standalone interventions, which could inform more comprehensive and personalized headache management.

Strengths and limitations

This study is distinguished by several methodological and conceptual strengths. Foremost, it represents the first application of ML-NMA to systematically compare the efficacy of diverse exercise modalities for migraine. This advanced analytical framework is a key strength, as it enabled us to overcome a critical limitation of traditional meta-analyses: the inability to properly account for statistical dependencies arising from studies with multiple treatment arms, outcomes, or follow-up times. Furthermore, the Bayesian nature of the framework allowed for the integration of prior information, enhancing the stability and precision of our estimates. Finally, and of significant clinical relevance, our pioneering exploration of dose–response relationships provides the first quantitative basis for optimizing exercise prescription, moving beyond simple efficacy comparisons to offer actionable guidance.

Notwithstanding its strengths, this analysis is subject to several limitations that warrant consideration. The conclusions of this synthesis are constrained by the low to very low certainty of the primary evidence, a reflection of endemic methodological weaknesses in exercise research. Advancing the field requires a concerted effort to improve primary study quality. Key priorities should include: (1) mitigating bias through the use of active comparators in non-blinded trials; (2) enhancing reproducibility and comparability by standardizing protocols and reporting (e.g., via CERT guidelines); and (3) generating definitive evidence through large, adequately powered, multi-center RCTs. While our sophisticated analytical methods provide the most robust synthesis possible from current data, true progress is contingent upon the methodological rigor of future primary studies. Second, our findings may be compromised by publication bias, for which funnel plot asymmetry provided suggestive evidence. Addressing this systemic challenge requires a field-wide commitment to research transparency, primarily through the universal adoption of prospective trial registration and the mandatory reporting of all outcomes. Third, some interventions were represented by a limited number of studies. While the ML-NMA methodology was specifically employed to bolster statistical power by “borrowing strength” across the network, the confidence in estimates for these sparsely studied interventions is necessarily reduced. Consequently, these particular findings should be considered preliminary. Finally, the marked female predominance within the study population, though consistent with migraine epidemiology, curtails the external validity of our conclusions for men. This highlights a critical knowledge gap and mandates future research designed to elucidate sex-dimorphic responses to exercise interventions.

Conclusion

This ML-NMA establishes a preliminary, evidence-based framework for exercise prescription in migraine management. Our analysis demonstrates that combined aerobic and resistance training, resistance exercise, tai chi, and yoga can yield clinically significant benefits. These effects are most pronounced when interventions adhere to a therapeutic window characterized by a weekly volume of 300–600 MET-min, an intensity of 4.5–5.5 METs, and a duration of 8–10 weeks. Therefore, these findings should not be interpreted as definitive clinical guidelines, but rather as an essential blueprint to inform the design of future high-quality randomized trials aimed at establishing robust exercise prescriptions for migraine.

Supplemental Information

Supplemental Information 1 Raw data

Supplemental Information 2 PRISMA checklist

Supplemental Information 3 Appendix

Additional Information and Declarations

Competing Interests

Author Contributions

Data Availability

The authors declare there are no competing interests.

Jingyi Xie conceived and designed the experiments, performed the experiments, analyzed the data, prepared figures and/or tables, authored or reviewed drafts of the article, and approved the final draft.

Yupeng Lin conceived and designed the experiments, performed the experiments, analyzed the data, prepared figures and/or tables, and approved the final draft.

Bin Wang conceived and designed the experiments, authored or reviewed drafts of the article, and approved the final draft.

The following information was supplied regarding data availability:

The raw data are available in the Supplemental File.

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
