# Peer review of "Efficacy and optimal dosage of various exercises for migraine: a multilevel network and dose-response meta-analysis"

_PeerJ, doi:10.7717/peerj.20254_

## Round 0.1 · original submission · Major Revisions

Thank you for your submission. Both reviewers find the topic timely and the methodology rigorous. However, revision is required to strengthen framing and clinical applicability. Specifically:
1) Expand discussion of generalisability (gender imbalance, sample characteristics).
2) Address publication bias more fully, including possible mitigation strategies.
3) Clarify the dose-response findings, especially why higher doses may not yield better outcomes.
4) Refine introduction and conclusions to sharpen the take-home message.
5) Add a section on the frequent coexistence of tension-type headache and migraine, and discuss potential benefits of combined exercise and CBT approaches as future directions.

These revisions will enhance the manuscript’s impact and clinical relevance.

·

Basic reporting

This manuscript presents a compelling and ambitious attempt to synthesize evidence on the efficacy of exercise interventions for migraine management, utilizing a sophisticated Multilevel Network Meta-Analysis (ML-NMA) paired with a dose-response analysis. The authors’ focus on comparing various exercise modalities—aerobic+resistance, resistance alone, Tai Chi, Yoga, and others—while also exploring optimal dosage parameters is a valuable contribution, particularly given the rising global burden of migraine and the growing interest in non-pharmacological treatments. The study’s strength lies in its methodological rigor, leveraging the ML-NMA framework to handle complex, hierarchically structured data, which offers a clear advantage over traditional meta-analytic approaches. The dose-response findings, pinpointing 300-600 MET-min/week at 4.5-5.5 METs over 8-10 weeks as a practical therapeutic window, provide actionable insights for clinicians, even if tempered by the low quality of evidence. The inclusion of diverse exercise types and the effort to quantify their effects with precision, as seen in the detailed SUCRA rankings and credible intervals, adds depth to the analysis, making it a meaningful step forward in understanding exercise as a migraine intervention. These findings align with and extend prior research, such as Woldeamanuel and Oliveira (2022), which found strength training to have the highest efficacy (MD = -3.55) in reducing monthly migraine frequency, followed by high-intensity aerobic exercise. The dose-response analysis here adds valuable nuance, suggesting a non-linear relationship where moderate doses balance efficacy and statistical certainty, offering clinicians actionable guidance.

However, the manuscript is not without flaws that warrant careful consideration. The overall quality of evidence, rated as very low to low via the CINeMA framework, casts a shadow over the findings’ reliability, a point the authors rightly emphasize but could explore further in the discussion to guide future research. The heavy reliance on female participants (80% of the sample) raises questions about generalizability, particularly for male patients, and this limitation feels underexplored. Publication bias, suggested by funnel plot asymmetry and Egger’s test, is another concern that could inflate effect sizes, yet the manuscript handles this transparently. The small number of studies for some interventions—like aerobic+resistance (k=1) and Tai Chi (k=1)—also weakens the robustness of those specific comparisons, which the authors acknowledge but could address more explicitly by discussing the feasibility of future trials to bolster these sparse data points. Additionally, while the dose-response analysis is a highlight, the non-linear relationship and its implications for clinical practice could be unpacked further, perhaps with more context on why higher doses might not yield better outcomes.

Overall, the manuscript is a thoughtful and well-executed effort that pushes the field forward but is constrained by the limitations of its source material. It offers a solid foundation for evidence-based recommendations while candidly highlighting the need for higher-quality, more diverse studies. I recommend acceptance with minor revisions to deepen the discussion of generalizability, publication bias mitigation, and the practical implications of the dose-response findings. This would enhance the manuscript’s utility for both researchers and clinicians seeking to translate these insights into practice.

Experimental design

details as above

Validity of the findings

details as above

Additional comments

details as above

Reviewer 2 ·

Basic reporting

Physical exercise has always been a controversial point in the comprehensive management of migraine. While high-energy anaerobic activities are considered a risk, low-energy aerobic exercise has long been recommended. The latter often contributes to the patient's physical and mental relaxation, particularly in chronic cases, where baseline performance is significantly reduced, negatively impacting overall quality of life. This can enhance the effectiveness of pharmacological therapy, both preventive and acute.

Experimental design

The systematic review with meta-analysis presented by the Authors aligns with the scientific evidence, which—though of low quality, as rightly noted in the Author's conclusions—holds significant presence in the literature and medical-scientific communication overall, especially on SoMe (Social Media).

Validity of the findings

Overall, the review is of good quality, thoroughly exploring all aspects of the chosen topic, with strong linguistic quality and appropriate methodology. In conclusion, aside from the suggestions listed earlier, the basic reporting is of broad and cross-disciplinary interes* and within the journal's scope, offering a well-developed presentation and supported arguments that meet the goals outlined in the Introduction. However, the conclusions and introduction should be refined according to the notes expressed earlier.

Additional comments

A concluding section on the frequent coexistence of tension-type headache and migraine, along with the potential benefits of combined physical exercise and CBT interventions, could further enhance the review by offering future research and clinical directions.

---

## Round 0.2 · accepted · Accept

Thank you for your careful and thorough revisions. You have addressed the reviewers’ comments very well, strengthening the clarity and interpretation of your work. The manuscript is now much improved and suitable for publication. We are pleased to accept your article for publication in its current form.

·

Basic reporting

Improved

Experimental design

Improved

Validity of the findings

Improved

Additional comments

None

Reviewer 2 ·

Basic reporting

The Authors addressed the raised concerns

Experimental design

The Authors addressed the raised concerns

Validity of the findings

The Authors addressed the raised concerns

Additional comments

The Authors addressed the raised concerns.